# Pulmonary Hypertension and the Gut Microbiome

**DOI:** 10.3390/biomedicines12010169

**Published:** 2024-01-12

**Authors:** Thomas Mason, Bhashkar Mukherjee, Philip Marino

**Affiliations:** 1Lane Fox Respiratory Service, Guy’s & St Thomas’ Hospital NHS Foundation Trust, London SE1 7EH, UK; 2National Pulmonary Hypertension Service, Royal Brompton Hospital, London SW3 6NP, UK

**Keywords:** pulmonary hypertension, gut microbiome, dysbiosis

## Abstract

The gut microbiome and its associated metabolites are integral to the maintenance of gut integrity and function. There is increasing evidence that its alteration, referred to as dysbiosis, is involved in the development of a systemic conditions such as cardiovascular disease (e.g., systemic hypertension, atherosclerosis). Pulmonary hypertension (PH) is a condition characterised by progressive remodelling and vasoconstriction of the pulmonary circulation, ultimately leading to right ventricular failure and premature mortality if untreated. Initial studies have suggested a possible association between dysbiosis of the microbiome and the development of PH. The aim of this article is to review the current experimental and clinical data with respect to the potential interaction between the gut microbiome and the pathophysiology of pulmonary hypertension. It will also highlight possible new therapeutic targets that may provide future therapies.

## 1. Introduction

Pulmonary hypertension (PH) is a progressive condition characterised by vasoconstriction and remodelling of the pulmonary vasculature, resulting in raised pulmonary pressures, increased right ventricular (RV) afterload and ultimately in RV failure. It is associated with significant morbidity and mortality. All currently approved treatments for PAH target mediators of pulmonary vasodilation via various pathways, aiming to reduce pulmonary vascular resistance (PVR) and right ventricular afterload.

PH is defined a mean pulmonary arterial pressure (mPAP) of >20 mmHg at right heart catheterisation. It is further characterised via measurement of the pulmonary capillary wedge pressure (PCWP, an estimate of left atrial filling pressures), which distinguishes precapillary pulmonary hypertension (low PCWP, ≤15 mmHg) from postcapillary pulmonary hypertension (high PCWP, >15 mmHg). Precapillary pulmonary hypertension suggests an increased PVR (>2 Wood units), whereas postcapillary pulmonary hypertension is a reflection of high left-sided filling pressures secondary to established left heart disease [1]. The causes of PH are summarised in Table 1. 

Pulmonary vasodilators are the mainstay of therapy for PAH [1]. These aim to reverse vasoconstriction, but do not significantly modulate the underlying disease process. There are three pathways that are targeted by PH therapies. Firstly, in the nitric oxide (NO)/soluble guanylate cyclase (sGC)/cyclic guanosine monophosphate (cGMP) pathway, NO stimulates sGC resulting in the production of cGMP in vascular smooth muscle cells, causing an influx of calcium ions and relaxation of the smooth muscle. Phosphodiesterase 5 (PDE5) inhibitors (e.g., sildenafil, tadalafil) reduce the degradation of cGMP, thereby promoting vasodilation. Riociguat also acts on this pathway by stimulating sGC. Secondly, in the endothelin pathway, endothelin-1 binds to endothelin receptors on smooth muscle cells causing vasoconstriction. Endothelin receptor antagonists (ERAs, e.g., bosentan, ambrisentan, macitentan) inhibit this pathway, thereby promoting vasodilation. Finally, prostacyclin analogues (e.g., iloprost, treprostinol) and prostacyclin agonists (e.g., selexipag) promote an increase in intracellular cyclic adenosine monophosphate (cAMP) in smooth muscle cells, causing vasodilation [1]. In PAH secondary to known associated factors (e.g., methamphetamine abuse, HIV, schistosomiasis) optimal treatment of the underlying cause is also necessary. The development of these therapies has led to further research in additional pathways that may be involved in the aetiology and progression of pulmonary hypertension.

The gut microbiome (GM) in humans is estimated to comprise 10^14^ microorganisms, predominantly anaerobic bacteria [2]. It has a role in many physiological processes, including gut mucosal integrity, vitamin synthesis, immunity, drug metabolism and energy metabolism. Each individual’s GM is unique, but can be altered by external factors such as diet [3], drugs [4] (particularly antibiotic use) and lifestyle.

The role of the GM in human health and disease is being increasingly recognised, particularly in cardiovascular disease. There is evidence that GM composition and metabolic activity are associated with the development of systemic hypertension, coronary artery disease/atherosclerosis and heart failure [5]. Abnormal GM (dysbiosis) causes translocation of bacterial metabolic products, consequent endovascular activation of macrophages and inflammation known to be associated with the development of several cardiovascular diseases including systemic hypertension, atherosclerosis and coronary artery disease [6,7,8]. In addition, bacterial DNA from microorganisms found in the oral cavity and GI tract have been identified in atherosclerotic plaques, although it is unclear whether this is causative or represents colonisation [9,10].

PAH is characterised by remodelling of the endovascular endothelium; however, the underlying trigger for this process is unknown in a high proportion of cases, hence the label of idiopathic PAH. It is hypothesized that, as with other cardiovascular diseases, the GM may play a role in the underlying pathobiology of PAH. Several conditions which are known to cause PAH, such as schistosomiasis, HIV infection, methamphetamine use and portal hypertension have been shown to have effects on the GM. For example, schistosomiasis infection (which is a leading cause of PH globally) causes irreversible changes in GM composition [11] with a reduction in GM diversity and it has recently been established that individuals who abuse methamphetamines have reduced GM diversity [12]. HIV infection is also associated with changes in GM dysbiosis, and it is hypothesised that this may result in chronic inflammation in patients with HIV [13].

The potential role of the GM in the pathophysiology of PAH is an exciting area which is not currently well understood. Greater understanding of the molecular mechanisms of PAH pathophysiology may ultimately result in therapeutic opportunities. In addition, clear characterization of the GM changes seen in PAH may have a future role in early diagnosis, which is important given the high morbidity and mortality associated with PAH.

## 2. Proposed Molecular Mechanisms

Figure 1 summaries the proposed interactions between the GM and the pathophysiology of PAH. Remodelling of the pulmonary vascular endothelium is characteristic of PAH and it is recognised that inflammation is a driver of this process, with Savai et al. demonstrating an increase in perivascular inflammatory cells in PAH patients [14]. A stable GM is essential in maintaining an intact gut epithelial barrier [15]. Alterations in GM are associated with increased bacterial translocation resulting in the entry of endotoxins such as lipopolysaccharides (LPS) into the circulation. LPS is a component of the cell walls of Gram-negative bacteria and as a pathogen-associated molecular pattern (PAMP), it is recognised by ancient pattern recognition receptors (PRRs) and is a strong stimulator of innate immune responses. The major PRR which interacts with LPS is toll-like receptor 4 (TLR4). TLR4 is expressed in lung endothelial cells and has been implicated in pulmonary vascular remodelling in PAH [16].

TLR4 activation results in downstream immune activation and cytokine release via the NF-κB pathway. In bone morphogenic protein receptor 2 (BMPR2) knockout mice treated with LPS there was an increase in TLR4 expression in pulmonary artery smooth muscle cells and a resultant increase in interleukin (IL)-6 and IL-8. This is also seen in patients with BMPR2 mutations [17]. Another ligand for TLR4 is HMGB1, a DAMP (damage-associated molecular pattern). HMGB1 is increased in both plexiform lesions of PAH patients and in chronically hypoxic rats. Stimulation of the HMGB1/TLR4 axis in hypoxic rats caused a decline in BMPR2, which is a well-established causative pathway in PAH [18].

TLR4-mediated signalling is also implicated in the regulation of vascular endothelial growth factors which are involved in endothelial cell proliferation [19] and in endothelial-to-mesenchymal transition (EndoMT). EndoMT is the process by which endothelial cells transition into vascular smooth muscle cells or cardiac fibroblasts during embryological development and has been observed in murine models of PAH [20]. Mice treated with LPS were found to have increased EndoMT [21]. However, the mechanism is clearly complicated as conversely, TLR4 knockout mice spontaneously develop pulmonary hypertension and increased pulmonary artery wall thickness. Furthermore, hypoxia was shown to downregulate TLR4 expression [22].

In addition to its role in maintaining the gut epithelium, the GM is also involved in the metabolism of indigestible fibre from the gastrointestinal tract. This process results in GM-derived metabolites such as short-chain fatty acids (SCFA) [23], trimethylamine N-oxide (TMAO) and Serotonin (5-HT). SCFA include acetic acid, propionic acid and butyric acid. Diet and the diversity of the GM both influence SCFA production. SFCAs are known to modulate immune responses via several mechanisms, including inhibition of histone deacetylases and activation of G-protein coupled receptors (GPCR). Inhibition of histone deactylase promotes differentiation of regulatory T cells (T-reg) which are anti-inflammatory mediators. SCFA activates a family of GPCR known as FFAR (free fatty acid receptors), which are found on many immune cells including macrophages and neutrophils [24]. GPCR43 (FFAR2) knockout mice showed exaggerated or unresolved inflammation in murine models of arthritis, asthma and colitis with increased production of pro-inflammatory cytokines [25].

Certain bacteria of the GM metabolise nutrients such as betaine and L-carnitine into trimethylamine, which is then oxidised in the liver to TMAO. Because of this, the composition of the GM dictates TMAO production. TMAO has been implicated as a risk factor for several cardiovascular diseases, particularly atherosclerosis [6]. Raised TMAO levels were also seen in mice with hypoxia-induced and monocrotaline-induced PH. Treatment of these mice with DMB (3,3-dimethyl-1-butanol, an inhibitor of TMAO synthesis) resulted in a reduction in pulmonary pressures and reduced pulmonary vascular thickness on histological analysis [26]. These data suggest that TMAO is linked to worsened PAH and may be a target for future treatments. A meta-analysis demonstrated that raised serum TMAO levels was associated with increased mortality and major adverse cardiovascular events in patients with heart failure [27].

The role of serotonin (5-HT) is well established in the pathophysiology of PAH. Serotonin is produced in endothelial cells by TPH1 (tryptophan hydroxylase 1) and expression of TPH1 in endothelial cells is increased in patients with PAH. Serotonin acts on nearby pulmonary artery smooth muscle cells (PASMCs) resulting in contraction (causing vasoconstriction) and proliferation. Serotonin also has an inhibitory effect on the BMP2R pathway [28]. The serotonin transporter (SERT) facilitates entry of serotonin into PASMCs and is also implicated in the pathogenesis of PAH. Overexpression of SERT results in PH in mice. The majority of 5-HT is synthesised in the gut and the GM is involved in regulation of 5-HT metabolism in the bowel [29] via SCFAs stimulating enterochromafin cells to increase transcription of TPH1 [30].

In addition to these molecular mechanisms, untreated PAH results in right ventricular (RV) dysfunction, causing venous congestion leading to reduced bowel perfusion and increased bacterial translocation of inflammatory mediators [31]. This may result in positive feedback and additional insult to the pulmonary vasculature. Gut-derived pathogens may translocate into the lung vasculature via lymphatic circulation, enlarged hepatic sinusoids or via portocollateral circulation.

## 3. Clinical Evidence for GM Involvement in PAH Pathogenesis

### 3.1. Distinct GM Patterns Found in PH

The typical spectrum of bacteria in healthy guts is well established, with *Firmicutes* and *Bacteroides* making up the dominant species [32]. The GM of PAH patients is distinct from healthy subjects. Kim et al. compared the faecal microbiome of 18 PAH patients to that of 13 reference subjects using shotgun metagenomics. The GM of PAH patients formed a distinct cluster of bacterial taxa with reduced alpha diversity, richness and evenness, with some typical changes, such as an increased *Firmicutes* to *Bacteroides* ratio (F/N ratio), typifying the gut dysbiosis seen in PAH. They also investigated the role of the gut virome and found that viral gene marker profiles were different between PAH and healthy cohorts [33]. *Lactococcal* phages were enriched within the reference cohort, while *Enterococcal* phages were enriched within the PAH samples, hinting at the potential role of the gut virome in regulating gut microbiome [33] (Table 2).

**Table 2 biomedicines-12-00169-t002:** Summary of GM changes in PH and their proposed role in PH pathobiology [31,32,33,34,35].

Changes in Microbiome	Functional Relevance in PH
Associated with downregulation of gut tight junction proteins -*Coriobacteriales* (*Collinsella*) Associated with preserved gut integrity: -Butyrate and proprionate-producing species -*Eg Roseburia* Associated with increased permeability -*Faecalibacteria* *A. muciniphila* *Coprococcus*, *Butyrivibrio*, *Lachnospiraceae*, *Eubacterium*, *Clostridia*	Gut barrier function
*Actinobacteria phylum*, (specifically *Bifidobacterium*) *B. crossotus*, *B. cellulosilyticus*, *E. siraeum*, *B. vulgatus*, *A. muciniphila*	Acetate synthesis -Increased in PAH
*Coprococcus*, *Butyrivibrio*, *Lachnospiraceae*, *Eubacterium*, *Clostridia* *Faecalibacterium prausnitzii*, *Ruminococcus (R.) bromii*, *Roseburia* *Bacteroides*	Butyrate synthesis -Reduced in PAH -Associated with reduced gut barrier function TMAO synthesis increase in PAH
*Akkermansia* and *Bacteroides*	Proprionate synthesis reduced in PAH
*lautia*, *Bifidobacteria*	Arginine metabolism -associated with PAH pathogenesis. -Bacterial ornithine transcarbamylase reduces arginine.
*Coriobacteriales (Collinsella)*	Miscellaneous cytokines increased epithelial IL-17
*Clostridium*, *Prevotella*, *aerofaciens*, *Clostridium*, *Staphylococcus*, *Streptococcus*, *Citrobacter*, *Coriobacteriales (Collinsella)*	TMA/TMAO synthesis increased in PAH
*Streptococcus*, *Coprococcus*	Tryptophan synthesis -Increased gut and circulating serotonin, Associated with PAH pathogenesis
*Coriobacteriales (Collinsella)*	Proline metabolism: -Increased in PAH, Associated with increased TMAO production.
*B. intestinihominis*	Purine metabolism -xanthine oxidase, purine nucleosidase Related to urate synthese, associated with PAH severity and prognosis

Similarly, an observational study by Ikubo et al. demonstrated similar findings in the relationship between gut dysbiosis and inflammation in CTEPH patients with decreased bacterial alpha diversity (the mean species diversity at a local scale) [34] (Table 2).

Faecal samples from a cohort of 73 PAH patients were analysed and compared to 15 family members and 39 healthy controls. While the GM profile in PAH patients was distinct and less diverse than in unrelated healthy subjects, they noted the difference in GM profile differed less between PAH and family than with controls [36], although whether this reflects cohabitation or inheritable factors remained unclear. It is of pathophysiological interest that there was an association between gut dysbiosis and pulmonary vascular disease but not with RV dysfunction (Table 2).

While the above studies consider the GM, Zhang et al. investigated the role of the airway microbiome in PH patients. In this study pharyngeal samples from patients with all PH groups were compared with those from control subjects. Once again, clearly distinct communities of bacteria were observed in PH patients compared to controls, with reduced community diversity in the upper respiratory tract of the PH cohort and a higher proportion of *Firmicutes* [37] (Table 2).

### 3.2. Microbiome Is Predictive of PAH

Moreover, Kim et al. found that this change in bacterial profile and the increased presence of specific bacterial species in their cohort predicted the presence or absence of PAH in subjects with 83% accuracy [33]. Similar modelling by Ikubo et al. also allowed robust prediction of CTEPH patients (with 80.3% accuracy) in their study [34].

## 4. Metabolic Features Associated with Postulated Disease Mechanisms

### 4.1. Altered Metabolic Profiles in PH

The PAH cohort studied by Kim et al. showed a profile of specific GM composition with decreased bacterial taxa associated with processes conferring gut health, such as polysaccharide fermentation and SCFA production, immune regulation and gut barrier fortification (Table 2). Similarly, Moutsoglou et al. showed that the GM in PAH patients were also enriched with respect to bacterial species associated with inflammation and proinflammatory microbial metabolites such as TMA [36] (Table 3).

**Table 3 biomedicines-12-00169-t003:** Microbiome composition and associated metabolic features in patients with pulmonary hypertension.

		Microbiome/Metabolic Findings in PH	Notes on Microbiome
Kim et al. (2020) [33]	18 group 1 PAH. 13 control subjects	↑ TMA/TMAO producing species ↑ acetate-producing species ↓ butyrate-producing species ↓ proprionate-producing species ↑ metabolites: Arginine, proline, ornithine, purine, urate	↓ alpha diversity
Zhang et al. (2020) [37]	118 PH. patients 79 control subjects	↑ richness of airways microbiota ↓ diversity of airways microbiota	↓ alpha diversity Key phylla: -*Firmicutes*, *Bacteroidetes*, *Proteo-bacteria*, *Actinobacteria*, *Fuso-bacteria* and *Saccharibacteria*
Ikubo et al. (2022) [34]	11 CTEPH patients 22 control subjects	↑ cytokines: -TNF-α, IL-6, IL-8 and MIP-1α ↑ endotoxin	↓ bacteria: -*Faecalibacterium* -*Roseburia* -*Fusicatenibacter*
Yang et al. (2022) [38]	163 PH patients (all groups)	↑ TMAO level -associated with severe PH and worse prognosis	
Moutsoglou et al. (2023) [36]	73 PAH patients 15 family controls 39 unrelated controls	↓ diversity of microbiota -Less microbiota difference between PAH and healthy family members ↓ SCFA producing species ↓ secondary bile acids ↑ TMA/TMAO producing species Evidence of gut permeability -↑ circulating Caludin-3 ↑ inflammatory cytokines (e.g., IL-6) No increase in CD14 or endotoxin	↑ proinflammatory species: -*Bacteroides thetaiotaomicron*, *Parabacteroides distasonis* and *Bacteroides vulgatus* ↓ anti-inflammatory species: -*Butyrivibrio* sp. and *Bifidobacterium angulatum*

### 4.2. Dysbiosis Associated with Gut Permeability, Translocation and Inflammation

GM imbalances in humans (as well as in animal models) allow translocation of bacterial components through the gut epithelial barrier into the bloodstream. This results in inflammation through macrophage activation, foam cell formation, cytokine release and abnormal platelet aggregation [7,8,39].

Increased gut permeability is seen in patients with PAH [40,41] as evidence by elevated markers such as endotoxin and CD14 [36,42]. Relative deficiency of anti-inflammatory bacterial communities, such as *Faecalibacteria*, is associated with increased gut permeability [42], whereas *Roseburia*, which promotes butyrate production, is associated with an anti-inflammatory milieu with increased activation of T-regs and preserved gut barrier function [43]. Kim et al. found a decrease in beneficial SCFA-producing bacteria in their PAH cohort, causing reduction in butyrate and propionate, which could potentially modify gut epithelial cells resulting in increased gut permeability [33].

Amongst the increased bacteria identified within the PAH cohort studied by Kim et al. were *Collinsella*, which may promote increased gut permeability via downregulation of gut tight junction proteins and increased epithelial IL-17A, as well as increased proline biosynthesis and TMA/TMAO production.

The increased levels of circulating Claudin-3 in PAH patients, noted by Moutsoglou et al. has been associated with increased gut permeability but as with previous studies, whether this is caused by or the cause of PAH and RV dysfunction cannot yet be ascertained [36]. It is known that venous congestion and reduced intestinal blood flow in heart failure can itself cause impaired gut barrier function and disruption of the GM [35,44]. There are also intriguing findings such as similar dysbiosis associated with increased gut permeability without the presence of raised IL-6 levels and the development of PAH in family members of PAH patients [36].

Pro-inflammatory cytokines such as IL-6 have been found to be elevated in PAH groups compared to controls [36] and have been shown to be correlated to increases in pulmonary vascular resistance [45]. Circulating IL-8 has been linked both with residual CTEPH after endarterectomy and poor prognosis in CTEPH patients [46].

Chronic inflammation is also thought to be important in the pathogenesis of CTEPH although the underlying pathogenesis has thought to differ from PAH [47,48]. Ikubo et al. (2022) showed that the typical accumulation of key macrophage-activated inflammatory markers in the PAH cohort, including tumour necrosis factor-alpha (TNF-alpha) which affects coagulation via the elevation of mononuclear tissue factor levels in CTEPH [45]. IL-6 [49], IL-9 and macrophage-inflammatory protein (MIP)-1-alpha [50], were positively correlated with increases in LPS levels, and increased in CTEPH patients compared to healthy controls [34].

### 4.3. Lipopolysaccharides

Increased LPS has been noted in PAH and CTEPH patients [33] and indicate possible increased gut permeability and translocation from the gut lumen as a source of endotoxin [51], which is known to cause macrophage activation and promote a cascade of pro-inflammatory processes associated with PAH pathogenesis [52].

The observed relative reductions in *Faecalibacterium*, *Roseburia* and *Fusicatenibacter* in CTEPH patients [34], also associated with reductions in endotoxin, were potential causes of increased gut inflammation and permeability [42,53,54]. Thus, metabolic endotoxaemia due to translocation of endotoxin induced by gut dysbiosis, which is already implicated in chronic low-grade inflammation in other cardiovascular diseases [39], may have a role in the pathogenesis of CTEPH.

### 4.4. TMA/TMAO

In Kim et al.’s PAH cohort, there was enrichment of *Coriobacteriales* bacteria implicated in the production of the metabolites TMA and TMAO [33]. Such TMA/TMAO-associated bacteria were significantly increased in PAH patients whereas bacteria negatively correlated with TMA/TMAO were enriched in non-PAH healthy subjects.

Yang et al. studied a cohort of 163 inpatients with PH (groups 1 and 4) over a median of 1.3 years. They found elevated plasma TMAO levels were associated with severe PH and a worse composite outcome, including mortality, rehospitalisation and change in 6 min walk distance of more than 15%. Clinical parameters including exercise capacity, cardiac function and risk stratification all correlated inversely with TMAO levels and, intriguingly, response to treatment of PH was associated with changes in TMAO levels [38]. This reinforces its potential use as a biomarker in PH, as in other cardiac diseases [55,56], although the precise mechanism remains unclear and causality in clinical trials has not been demonstrated clearly. As with brain natriuretic peptide (BNP), it may reflect increased hydrostatic pressures due to venous congestion rather than directly promote development of PAH [56]. In addition, TMAO may be involved in the progression of PH by inducing inflammation and endothelial permeability [57].

TMAO levels are increased in patients with high- and intermediate-risk PAH but not low-risk PAH or controls [26]. Subsequent in vitro studies suggest the mechanism may involve inducing inflammation via macrophage stimulation rather than by direct action on pulmonary smooth muscle cells.

### 4.5. Short Chain Fatty Acids (SCFAs)

In the PAH cohort of patients studied by Kim et al., there was an increase in acetate-producing bacteria (particularly *Bifidobacterium*), compared to more butyrate and propionate-producing bacteria in the faecal microbiota of reference subjects [33]. On a phylogenetic level and in terms of relative abundance of species, Moutsoglou et al. showed PAH patients had fewer bacteria producing anti-inflammatory short-chain fatty acids (SCFAs) and correspondingly lower serum levels of circulating SCFAs and secondary bile acids [36].

Increased plasma arginase and consequent decrease in arginine bioavailability is a feature of PAH pathogenesis. Kim et al. reported increased arginine, proline and ornithine biosynthesis and interconversion in the PAH cohort microbiome [33] (Table 2), highlighting how increased ornithine transcarbamylase activity in PAH patients’ GM could account for the decreased availability of arginine and hence be important to PAH pathogenesis.

Given the known association of purine metabolism, and particularly urate, with PAH severity and mortality, Kim et al. also noted xanthine oxidase and purine nucleosidase were among purine-metabolising enzymes found to be enriched in the PAH cohort [33].

### 4.6. Serotonin

Given serotonin is implicated in the pathogenesis of PAH, it is noteworthy that bacterial tryoptophan biosynthesis and circulating serotonin is increased in PAH patients [33]. While there has been extensive preclinical data establishing the importance of serotonin and serotonin transporter (SERT) in the development of PAH, further clinical studies are necessary. Dexfenfluramine is a serotonergic drug which acts on SERT and has been established as a cause of PAH [28].

### 4.7. Respiratory Dysbiosis Implicated in PAH Pathogenesis

As in other respiratory conditions, a reduction in airway microbiota is not yet shown to be implicated as causing PH, although the microbiota imbalance are implicated in the pathophysiology [58,59,60]. Just as Kim et al. showed increased *Streptococcus* in the GM of PH patients, this has also been demonstrated in the oral microbiota of PH patients [37]. *Streptococcus* has been suggested to have a role both in airways dysregulation and in the development of PH by means of activation of PI3K and ERK signalling pathways which are involved in the proliferation of pulmonary artery smooth muscle cells. Enrichment of the lung microbiome may be associated with inflammation of Th17phenotype, which is also characteristic of inflammation commonly found in PH patients, with production of cytokines such as IL-17 [61,62]. Similar cytokine profiles have been associated with supraglottic predominant taxa such as *Prevotella* and *Veillonella*.

### 4.8. Summary

Most of the studies to date compare relatively small numbers of PH patients, providing interesting associations rather than clear causality. It is nonetheless interesting that the changes in SCFAs, endotoxins and inflammatory cytokines seen in PAH have been shown to be produced from the characteristic microbiome communities seen in PH patients.

Future microbiome–PAH studies in humans need to ensure matched cohorts, including age and dietetic factors, to assess PAH-associated heritable genetic mutations, and further evaluate the role of comorbidities and gender, given the relative female predominance of PAH.

## 5. Potential GM-Based Therapies in PAH

Our increasing understanding of the role of the GM and dysbiosis within the gut and respiratory tract could provide new treatments for PAH that may modify the underlying disease process. There is already evidence that modification of the GM may slow disease progression in several chronic conditions such as diabetes, atherosclerosis, systemic hypertension and some tumours. The main aim in each case is either altering the microbiome to an anti-inflammatory profile or to achieve restoration of the original bacterial profile. We will discuss the most current therapeutic targets below.

### 5.1. Probiotics

The use of live microorganisms to restore the GM and its integrity has been applied with some success in patients with luminal disorders such as irritable bowel syndrome [63]. An early clinical trial in male patients with stable angina showed a reduction in both systemic inflammation and vascular endothelial dysfunction by altering levels of SCFAs [64]. Certain organisms such as *Lactobacillus reuteri* have been found to reduce pro-inflammatory cytokine levels and inhibit TLR4 signalling through the NF-kB pathway, as well as modifying pathophysiological changes in animal models of PAH [65,66]. Studies have focused on probiotics from gut bacteria, but such immune-modulatory effects might also occur with probiotics derived from the respiratory tract. The combination of probiotics with current PAH therapies may provide more effective treatments with future research identifying which regimens are most effective and safe.

### 5.2. Prebiotics

An alternative approach to giving live microorganisms has been the use of prebiotics. These are complex molecules that may either modify the GM to promote the growth of beneficial bacteria (e.g., *Lactobacillus*) or can be utilised by normal host microorganisms to promote an anti-inflammatory profile and/or suppress vascular remodelling [67,68]. Prebiotics may include complex carbohydrates (oligo- or polysaccharides) and polyunsaturated fatty acids [69,70]. Such molecules have been shown to prevent progression in animal models of PAH [68,71] with the next step being clinical studies in PAH studies. This will establish which molecules and regimens are both efficacious and safe and may lead to combination therapy with probiotics for a greater synergistic effect, referred to as synbiotics.

### 5.3. Faecal Microbiota Transplantation (FMT)

The aim of FMT is to restore the GM and luminal integrity through the introduction of normal gut bacteria into the gastro-intestinal tract by transferring specially filtered stool from normal healthy donors. Animal studies have shown improved pulmonary haemodynamics and reduced remodelling in hypoxia-induced PAH in wild-type mice that received faecal matter from ACE2 knock-out mice [72]. This technique has already been applied therapeutically in humans in the treatment of recurrent *Clostridium difficile* infection [73] as well as being considered in the treatment of the metabolic syndrome [74]. This has stimulated interest in clinical trials in humans, with the approval last year of a phase 1 clinical trial in PAH patients. Further trials will help to establish its safety and which microbiota regimens are effective.

### 5.4. Antibiotics

Certain animal studies have attempted to eradicate bacteria that promote increased gut permeability and inflammation through the use of combination antibiotic therapy [75]. These regimens have used antibiotics with a relatively broad spectrum that have targeted these bacteria but may also eradicate non-pathogenic bacteria or those with anti-inflammatory profiles. This has understandably raised concerns of increasing bacterial resistance or colonisation by other organism (e.g., fungi) that could lead to significant sepsis caused by multi-drug resistant organisms. Consequently, this approach has not garnered wider interest as a possible intervention.

### 5.5. Dietary Modification

The benefits of a Mediterranean diet (high in olive oil but low in red meat and dairy) on reducing cardiovascular risk have been well documented. This had led some groups to explore the effects of specific diets on PAH in both animal models and human cohorts. For example, high-fat diets have been shown to worsen pulmonary pressures and cause RV hypertrophy in certain mouse models [76,77]. These diets are often rich in choline and L-carnitine that are precursors of TMAO that causes endothelial dysfunction through promoting inflammation and oxidative stress [78]. However, diets that are high in fibre and foods containing antioxidants may have protective effects on both the gut microenvironment and cardiovascular risk through modulating the GM and its metabolites (principally SCFAs), thereby reducing inflammation and endothelial damage [79,80].

Intermittent fasting regimens have shown potential benefits in terms of modifying disease progression and mortality in animal models of certain chronic diseases [81]. Similar outcomes have been demonstrated in animal models of PAH with improved RV function (through reduced RV hypertrophy and fibrosis), restored gut integrity and enhanced growth of bacteria (e.g., *Lactobacillus*) with anti-inflammatory properties, leading to reduced mortality [82].

There are currently clinical trials in humans investigating the utility of these various dietary regimens in cardiovascular and metabolic disorders. This should hopefully prompt similar studies in PAH patients.

### 5.6. Oral Hypoglycaemic Agents

There is evidence of reduced exercise capacity in PAH patients with associated increased insulin resistance or impaired secretion, whilst reduced survival was observed in those patients with coexistent diabetes [83,84]. These observations promoted interest in assessing the potential efficacy of oral hypoglycaemic agents as adjunct therapies in PAH. Metformin has been shown to prevent adverse pulmonary haemodynamics and RV remodelling in mice on high-fat diets that had undergone PA banding [85]. Newer hypoglycaemic agents such as the Sodium-Glucose Transporter Type 2 (SGT2) agonists or Glucagon-Like Peptide-1 (GLP-1) agonists may potentially offer similar or greater benefits in PAH patients.

### 5.7. Mesenchymal Stem Cell (MSC) Therapy

Mesenchymal stem cells (MSC) may provide another treatment modality with a direct effect on the pulmonary vasculature and right heart, as well as altering GM favourably. Animal studies have showed a reduction in pulmonary pressures and RV remodelling through increased capillary density and by suppressing cardiac hypertrophy and fibrosis [86]. Similar improvements in pulmonary haemodynamics have been demonstrated in murine models of hypoxia-induced PAH with MSC, associated with a reversal and restoration of the GM leading to a more anti-inflammatory profile [87]. Further research is clearly required to develop such therapies in patients with PAH.

### 5.8. Screening Tests

The alterations in the gut and respiratory microbiome that have been identified in PAH patients may provide a screening tool to detect dysbiosis and then start potential therapies as described earlier. For example, certain studies have shown enrichment of *Enterococcus* and depletion of *Lactobacillus* in the gut [33], with others showing a reduction in SCFA-producing bacteria in PAH patients [88]. Studies of the respiratory microbiome in PAH have identified an elevated growth of TMA/TMAO producing bacteria such as *Streptococcus* or an increase in the F/B ratio [33,37]. The development of any screening test would need to consider the influence of variables such as age, PH group, comorbidities, medication (especially recent or prophylactic antibiotics), diet and geographical area and must be validated with respect to these factors.

## 6. Conclusions

Animal and human studies have identified changes in the GM that are associated with pro-inflammatory changes at these sites and in the pulmonary circulation that may be associated with development of PAH. Further studies are clearly needed to fully characterise the GM in PAH, as well as establish causality between gut dysbiosis and disease progression in PAH in humans. Several intrinsic and extrinsic factors, including genetics, nutritional habits, pharmacological treatments, ethnicity, age and sex, also influence the maintenance of GM in healthy individuals. This may then provide clinicians with potential new therapies that target dysbiosis and could be used in conjunction with current PAH therapies, after proper assessment in formal clinical trials.

## 7. Key Learning Points

Alterations in the GM (dysbiosis) have been found in patients with PAH and are associated with pro-inflammatory effects and increased gut permeability.These changes may be a contributing factor in the development and progression of PAH.Therapies (e.g., probiotics, FMT) that can potentially reverse dysbiosis might be used in the future with current PAH medications dependent on further clinical studies establishing safety and efficacy.Screening tools for dysbiosis may become available through sampling and characterisation of the gut and respiratory microbiome.

## Figures and Tables

**Figure 1 biomedicines-12-00169-f001:**
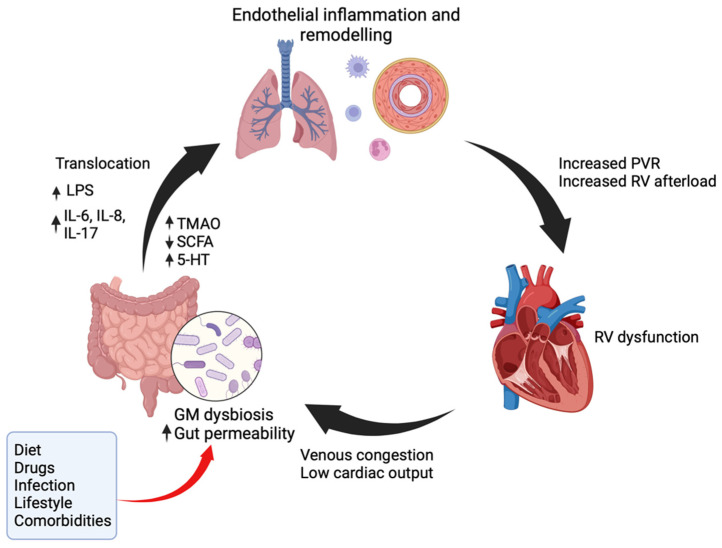
Schematic summarising the proposed interactions between the GM and the pathophysiology of PAH.

**Table 1 biomedicines-12-00169-t001:** Adapted from 2022 ESC/ERS Guidelines for the diagnosis and treatment of pulmonary hypertension [1].

Group	Description	Examples	Haemodynamic Profile
1	Pulmonary arterial hypertension	Idiopathic Heritable Drug induced Associated with: -Connective tissue disease -Congenital heart disease -HIV -Portal hypertension -Schistosomiasis Pulmonary veno-occlusive disease	Precapillary, raised PVR
2	PH secondary to left heart disease	LV systolic or diastolic dysfunction Valvular heart disease	Postcapillary, normal/raised PVR
3	PH secondary to chronic lung disease/hypoxia	Obstructive or restrictive lung disease Sleep-disordered breathing Altitude	Precapillary, raised PVR
4	PH associated with pulmonary artery obstructions	Chronic thromboembolic disease (CTEPH)	Precapillary, raised PVR
5	PH with unclear/ multi-factorial mechanisms	Systemic disorders (e.g., sarcoidosis) Chronic renal failure/haemodialysis	Pre-/postcapillary, normal/raised PVR

## Data Availability

Not applicable.

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
