# Peer review of "Pulmonary Hypertension and the Gut Microbiome"

_biomedicines, 2024, doi:10.3390/biomedicines12010169_

Round 1
Reviewer 1 Report
Comments and Suggestions for Authors
This is a very interesting review article on the interactions between gut microbiome and pulmonary hypertension.
Section 3.1: "Several studies have now demonstrated consistent changes in the microbiome of PH patients." This sentence needs to note references.
Author Contributions section should be filled.
Table 2: References need to be noted in this table.
Author Response
We appreciated your comments & have fully addressed them in the new version of the article.
Reviewer 2 Report
Comments and Suggestions for Authors
The authors of the manuscript analyzed and presented the current state of knowledge regarding dysbiosis and pulmonary hypertension. The work is interesting, but in my opinion it requires expanding several aspects.
- Intestinal dysbiosis in heart failure-modulation of dysbiosis may be considered as a potential therapeutic target. Similarly, oral microbial dysbiosis may play a role in cardiovascular diseases. Please discuss in more detail the importance of cardiovascular diseases in a separate section of the introduction.
- one of the basic therapeutic directions in PAH is the (multidirectional) influence on the guanylate cyclase and cGMP pathway. Soluble guanylyl cyclase activators are promising therapeutic option in the pharmacotherapy of heart failure and pulmonary hypertension. Carbon monoxide and nitric oxide are examples of the youngest class of transmitters, please describe in a separate chapter current modulation of guanylate cyclase pathway activity — mechanism and clinical implications, especially in the aspect of pulmonary hypertension.
- Artificial intelligence technologies are becoming increasingly important in cardiology. According to the authors, should the problem of dysbiosis be an element in risk stratification?
- the work is interesting primarily for practicing physicians, hence I am asking you to propose more decisive, practical suggestions in the conclusion.
Author Response
Thank you for your comments. After further discussion we have addressed them as follows :
1.The section on the role of oral dysbiosis on cardiovascular disease has been expanded.
2.We would have to respectfully disagree with the suggestion of including a chapter on the guanylate cyclase pathway activity. This is a well-established therapeutic pathway in PAH and already extensively documented/discussed in the PH literature. The remit of the review article was to discuss the potential relationship between the development and pathophysiology of PH with the host microbiome, not a review of the current therapeutics.
3.We have already outlined in section 6 the possibility of developing screening tests for dysbiosis in PH and using it to guide future therapy.
4.As mentioned throughout the article this area of research is still evolving, especially with respect to potential future diagnostics and therapeutics. Hence, the suggestions made are based upon this early data and not definitive recommendations for clinicians, pending further research.
Round 2
Reviewer 2 Report
Comments and Suggestions for Authors
The authors have partially introduced corrections to the work. I would like to return to the issue of presenting the basis of current pharmacotherapy in the manuscript, which the authors strongly opposed. In the introduction there is one sentence mentioning high mortality, in the further part we read that vasodilator therapies are used. in the next steps, the authors discuss their issues of the intestinal microbiome. In my opinion, such a presentation misleads the reader by suggesting that there are some treatment methods, but that intervention within the presented issue may be an appropriate option. Considering the role - the microbiome may be a supportive therapy due to the etiology of PAH. It may turn out that it will play a huge role in preventive activities and diagnostic procedures. The work avoids the issue of pathomechanism and pathways used in therapy (ET-antagonists, prostacyclin, PDE5 inhibitors or GC stimulators). In this way, the reader will not locate the problem of the intestinal microbiome within therapeutic processes.
I believe that it is necessary to add a sentence that emphasizes the importance of diagnosis and rapid initiation of treatment, introduces the pathways currently used in therapy and outlines the potential directions of the positive impact of the intestinal microbiome (here is a clear link to GC pathways). In conclusion, the role of a potentially supportive therapeutic option should be clearly outlined.
Author Response
'Thank you again for your comments. We have discussed the established therapeutic pathways that are currently utilised in the management of PAH as suggested, using it as a link to the Gut microbiome and the inflammatory processes that are increasingly thought to be involved in the aetiology and development of pulmonary hypertension.
We feel this has provided the appropriate balance to this review that was requested whilst simultaneously fulfilling the original remit given.'
Round 3
Reviewer 2 Report
Comments and Suggestions for Authors
I believe that, once amended, the manuscript may be considered for publication